# MicroRNA Modulation of Host Immune Response and Inflammation Triggered by *Helicobacter pylori*

**DOI:** 10.3390/ijms22031406

**Published:** 2021-01-30

**Authors:** Maria Oana Săsăran, Lorena Elena Meliț, Ecaterina Daniela Dobru

**Affiliations:** 1Department of Pediatrics III, “George Emil Palade” University of Medicine, Pharmacy, Sciences and Technology of Târgu Mureș, Gheorghe Marinescu Street no 38, 540136 Târgu Mureș, Romania; oanam93@yahoo.com; 2Department of Pediatrics I, “George Emil Palade” University of Medicine, Pharmacy, Sciences and Technol-ogy of Târgu Mureș, Gheorghe Marinescu Street no 38, 540136 Târgu Mureș, Romania; 3Department of Internal Medicine VII, “George Emil Palade” University of Medicine, Pharmacy, Sciences and Technology of Târgu Mureș, Gheorghe Marinescu Street no 38, 540136 Târgu Mureș, Romania; danidobru@gmail.com

**Keywords:** *H. pylori*, miRNA, TLR, host immune response, gastric inflammation

## Abstract

*Helicobacter pylori (H. pylori)* remains the most-researched etiological factor for gastric inflammation and malignancies. Its evolution towards gastric complications is dependent upon host immune response. Toll-like receptors (TLRs) recognize surface and molecular patterns of the bacterium, especially the lipopolysaccharide (LPS), and act upon pathways, which will finally lead to activation of the nuclear factor-kappa B (NF-kB), a transcription factor that stimulates release of inflammatory cytokines. MicroRNAs (MiRNAs) finely modulate TLR signaling, but their expression is also modulated by activation of NF-kB-dependent pathways. This review aims to focus upon several of the most researched miRNAs on this subject, with known implications in host immune responses caused by *H. pylori*, including let-7 family, miRNA-155, miRNA-146, miRNA-125, miRNA-21, and miRNA-221. TLR–LPS interactions and their afferent pathways are regulated by these miRNAs, which can be considered as a bridge, which connects gastric inflammation to pre-neoplastic and malignant lesions. Therefore, they could serve as potential non-invasive biomarkers, capable of discriminating *H. pylori* infection, as well as its associated complications. Given that data on this matter is limited in children, as well as for as significant number of miRNAs, future research has yet to clarify the exact involvement of these entities in the progression of *H. pylori*-associated gastric conditions.

## 1. Introduction

*Helicobacter pylori* (*H. pylori*) is a Gram-negative, urease producing, spiral-shaped bacterium, which infects more than half of the world’s population [1,2]. The colonization of the antrum and fundic gastric mucosa can cause significant lesions, ranging from acute, chronic inflammation to gastric cancer [3]. Classified by the World Health Organization (WHO) as a first line carcinogen, prolonged exposure of the gastric mucosa to *H. pylori* toxins leads to molecular changes, affecting cellular DNA and RNA [4]. Therefore, *H. pylori* is both indirectly and directly involved in carcinogenesis, by promoting an inflammatory environment inside the gastric epithelial cells and by inducing protein synthesis and genetic mutations [5]. As the bacterium is often acquired during childhood, host inflammatory response plays an important role in the progression of this infection [1]. *H. pylori* presents a lipopolysaccharide (LPS), virulence factors, and even the capacity of forming a biofilm, which helps in neutralizing immune response. Biofilm formation has been attributed to morphological changes (cell wall rearrangement), membrane vesicles secretion, and inter-microbial communication, which constitute virulence and adaptive responses of the pathogen [6,7]. Eradication of the infection is therefore essential for treating gastritis and preventing premalignant alterations of the gastric mucosa, although antibiotic resistance is increasing worldwide, enforcing continuous search for novel therapeutic approaches [8]. Nevertheless, standard eradication regimens recommended by current guidelines in treating *H. pylori* can also positively impact precancerous lesions, by improving glandular atrophy, as well as intestinal metaplasia, in a selected number of cases [9,10].

Innate immunity is a crucial component of the host’s immune system involved either in the promotion or in eradication of *H. pylori* infection. MicroRNAs (miRNAs) encompass small, non-coding fragments of RNAs, with dimensions ranging between 17 and 24 nt [11], and seem to play a major role in the host’s immune responses. These sequences regulate gene expression at a post-transcriptional level, being responsible for RNA silencing [11,12]. This function is achieved through attachment of miRNAs to two target regions of messenger RNA (mRNA): 3′-untranslated region (3′-UTR) or open reading frame [13]. As a result, microRNAs will provide a specific sequence of nucleotides, which will enable ribonucleoproteins to direct splicing, in a similar manner to small nuclear RNAs [14]. Endonucleolytic cleavage takes place, but sequences of recognition can casually cause mismatches [15]. A degree of complementarity does however influence RNA silencing, since high affinity of microRNAs for their target will cause mRNA degradation, whereas low levels of interactions will result in translational repression [16]. Still, miRNAs do not exercise a complete repression upon their targets, but rather decrease gene expression. Inhibition of protein synthesis is responsible for this effect, miRNAs acting as instruments, which can have a subtle, yet important effect on the proteome [17]. Being involved in the signaling of cell proliferation pathways, miRNAs can act on the cell cycle by regulating expression of various proteins responsible for suppression of cell cycle progression. Therefore, dysregulation of miRNAs might result in aberrant cell proliferation, an important hallmark of tumorigenesis [18]. Thus, it is unsurprising that even a single type of miRNA can produce contradictory effects in different organs and tissues, in relation to the level of targeted gene expression [19].

MiRNAs have a cellular source, their biogenesis implying multiple nuclear and cytoplasmatic maturation processes being eventually released in the extracellular compartment. Their extracellular delivery is ensured by exosomes, microvesicles, protein, and lipoprotein complexes. MiRNAs have been identified in numerous extracellular bio-fluids, including serum, plasma, urine, saliva, gastric juice, bronchial lavage, pleural peritoneal, or synovial fluids [20]. Therefore, quantifying their expression does not always imply the assessment of tissue biopsies, being also achieved using various bio-fluid samples.

The involvement of miRNAs in diverse biological processes has created the premises for multiple studies, some of these proposing them as highly sensitive and specific biomarkers [16]. Connection between miRNAs and inflammatory processes has been intensely studied recently, these entities emerging as key elements of cellular communication, with an active role in cellular proliferation and differentiation, autoimmunity, carcinogenesis, as well as innate immune responses. Pathogens trigger these responses, which represent the first defensive structure against pathogens and initiate inflammation after recognition of specific structures of the invading organisms [21]. MiRNAs are fine modulators of the immune response as a result to pathogen exposure, usually ensuring that the host will have a moderate, appropriate reaction. Apart from their implication in modulating inflammatory cascades, they are also involved in negative feedback mechanisms responsible for innate immune response [22]. These findings have opened new research gates for miRNA implications in the outcomes of multiple infectious agents, including *H. pylori*. This bacterium not only stimulates both innate and adaptive immune responses, but also alters the expression of various types of miRNAs, signaling certain pathways that enhance premalignant transformation of chronic inflammation [23]. 

This review aims to highlight the role of miRNAs on modulating host immune response, gastric inflammation, and progression of gastric lesions towards *H. pylori*-induced neoplastic conditions. Considering that the precise involvement of most miRNAs in regulation of these pathways is yet unknown, our intention is to focus upon several, most-studied miRNAs on this subject, in light of recent available data.

## 2. *Helicobacter pylori*, Virulence Factors, and Host Immune Response

Multiple specific properties of *H. pylori* promote its survival within the gastric acid and subsequent recruitment of inflammatory cells. Details of its structure, as well as its virulence factors are outlined in Figure 1. Its helicoidal shape is essential for efficient colonization of the gastric mucosa. An inner membrane protein, i.e., Csd7 plays a key role in both maintaining shape stability and regulating the cellular wall, by interacting with two other proteins: Csd2 protein and Csd1, a peptidoglycan endopeptidase [24]. This complex helicoidal structure has been considered a virulence factor by some authors [25]. The lipopolysaccharide (LPS) represents another important structure, situated on the surface of *H. pylori*, which promotes invasion, host interaction, and helps the bacterium escape protective mechanisms of the gastric mucosa [26]. HupA, the main enzyme of LPS, favors resistance to cationic antimicrobial peptides (CAMPs) through a currently unknown process, presumptively inducing changes in phospholipid synthesis. Survival of *H. pylori* is thus ensured, as well as its capacity to evade innate immune response [27]. Specific morphology and LPS act in conjunction with the *H. pylori* urease to ensure effective colonization. This enzyme is located inside the cytoplasm and facilitates survival of the bacterium in acid environment, by hydrolyzing urea into ammonia and carbonic acid. Therefore, a constantly alkaline pH is ensured inside the cytoplasm, the enzymatic activity increasing with decrease of external/environmental pH [28,29]. Furthermore, the production of ammonia disrupts tight epithelial junctions through accumulation of a soluble form of occludin [30]. Disruption of the gastric epithelial barrier, the first layer of protection against pathogens occurs as a result of cytotoxicity and impaired epithelial mitochondrial oxygenation, also caused by high levels of ammonia [29].

Alteration of the epithelial barrier will set off innate detection mechanisms, implying recognition of pathogen-specific molecular structures. Pattern-recognition molecules (PRMs), such as toll-like receptors (TLRs) can identify these components and trigger an innate immune response [31]. Thus, TLRs recognize pathogen-associated molecular patterns (PAMPs), particular surface or membrane components of *H. pylori*, as well as its nucleic acids [32,33]. Their main role is to activate intracellular signaling pathways, the most widely studied being MyD88- and Trif-mediated ones, which will commonly reach the same endpoint- activation of the nuclear factor-kappa B (NF-kB), a transcription factor which induces the release of inflammatory cytokines [34]. TLR2 and TLR4 have been proven to serve as specific ligands for *H. pylori* LPS, although they both recognize different LPS structures [32]. TLR2-LPS interaction will result in high levels of interleukin 8 (IL 8), as TLR2 is directly responsible for activating its promoter. LPS specifically binds to TLR4 on the mononuclear cell surface and stimulates release of other types of cytokines, IL-12 and IL-1β, which induce T helper (Th) 1 and Th17 differentiation, as well as IL-18, an inhibitor of IL-1β effects. Therefore, overexpression of IL-18 will suppress immune response and help the pathogen escape the action of T helper cells [6]. Contrariwise, increased expression of IL-1β will decrease the activity of the gastric proton pump (H+/K+-ATPase) which will lead in time to gastric atrophy [4]. Furthermore, TH17 stimulates release of IL-17A, which induces parietal cell apoptosis, and subsequent atrophic gastritis [35]. Thus, it is not unlikely for TLRs to be responsible for triggering immune and inflammatory pathways, which require a fine-tuning in order to prevent an excessive response.

Immune response is further modulated by other three major virulence factors: *H. pylori* neutrophil-activating protein (HP-NAP), cytotoxin-associated gene A (CagA) and the vacuolating cytotoxin (VacA) [2,36]. HP-NAP has gained its name due to its chemotactic properties for neutrophils, as well as its role in inducing release of reactive oxygen species (ROS), which damage epithelial cells [4,37]. A TLR2 agonist, HP-NAP is a mediator of both innate and adaptive immune response, by promoting TH1 cell differentiation, stimulating IL-12 and IL-23 release from neutrophils and monocytes and upregulating the major histocompatibility complex class II (MHC II) [36]. CagA is considered a major mediator of carcinogenesis, due to its peculiar characteristics of disrupting epithelial cell junctions, inhibiting the H+/K+-ATPase and of activating intracellular pathways that will finally determine cell proliferation and transformation [38,39,40]. Its presence has been reported in 60% of *H. pylori* strains, being considered a strong inducer of the NF-kB [39,41]. VacA alters the integrity of the epithelial barrier and impacts the β-catenin pathway, being able to transcribe oncogenes, such as cdx1 and cyclin D [2,42]. VacA can cause both apoptosis and inflammation through stimulating release of pro-inflammatory cytokines, such as IL-1β, IL-6, IL-10, and tumor necrosis factor α (TNFα) [43]. Unlike HP-NAP, VacA inhibits MHC class 2-dependent pathways and favors survival of *H. pylori* within macrophages by forming vesicular compartments inside them [44].

Neutrophil infiltration into lamina propria represents the first step towards development of inflammation. Urease and HP-NAP represent major chemotactic factors for neutrophils, whereas LPS and HP-NAP contribute to their activation. Furthermore, outer membrane protein Q of *H. pylori* promotes neutrophil survival [45]. Infection persistence will determine arousal of lymphocytes within the gastric mucosa, a hallmark of chronic inflammation. Therefore, lymphocytes are more often identified in adult gastric biopsy specimens than in those from children, concomitant presence of neutrophils suggesting an active, chronic gastritis [46]. Taking into account that most individuals acquire the infection during childhood [1], these differences in histological findings between adults and children further sustain that its persistence over time results in chronic changes of the gastric mucosa. According to the updated Sydney classification system, inflammatory infiltrates represent reliable indicators of gastritis and *H. pylori* infection severity [47]. Reproduction of an infection model on mice revealed that macrophage and neutrophil infiltration of the gastric mucosa occurs within two days after *H. pylori* inoculation. However, their number subsides gradually and increases again 2–3 weeks post-infection. Infiltration of the gastric mucosa by T lymphocytes is an essential part of the adaptive immune response, which occurs approximately 20 days after *H. pylori* inoculation. T cytotoxic cells (CD8+ lymphocytes) were significantly less abundant than T helper cells (CD4+ CD3+ lymphocytes), but both types were also present in the paragastric lymph nodes, together with macrophages and dendritic cells [48]. Therefore, a strong production of interferon γ (IFNγ) within lymph nodes, also reported by other studies is expectable [49]. A computational model, developed in order to predict mucosal immune response after *H. pylori*, sustains the essential implications of T helper cells (Th), Th1, and Th17 cells as the main agents that induce changes within the gastric mucosa [50]. Moreover, immunophenotyping of gastric lamina propria in *H. pylori*-infected mice showed a significant increase in IL-17A and IFNγ 30 to 60 days after bacterial inoculation. The mucosal damage progressed to day 60, all mice exhibiting microscopic changes characterized by elongation of gastric pits and basal glandular cell depletion [51].

Numerous pathways are involved in the pathogenesis of *H. pylori* infection. Moreover, from the virulence factors of the bacterium itself, innate and adaptative immune responses significantly contribute to the development of gastric histopathological changes. Thus, modulation of various pathways could represent the key for developing and preventing malignant transformation.

## 3. *Helicobacter pylori*—Driven Pathways of Gastric Carcinogenesis

Involvement of *H. pylori* in the development of gastric cancer represents an area of continuous research, with still incompletely described signaling pathways. Innate immune response seems to trigger multiple key effectors of gastric carcinogenesis, whereas virulence factors also play their roles in this process.

As previously described, activation of NF-kB-dependent pathways leads to release of various inflammatory cytokines. MD-2 protein facilitates recognition of LPS by TLR-4, interaction of these two entities resulting not only in the activation of NF-kB, but also in augmentation of IL-8 promoter function [52]. IL-8 can significantly contribute to gastric oncogenesis, as represented in Figure 2. IL-8 acts as a chemotactic agent for neutrophils, which correlate with gastritis severity and stimulate release of vascular endothelial growth factor (VEGF), as well as proliferation and migration of endothelial cells, subsequently promoting angiogenesis [53]. This chemotaxis is achieved via CD11b/CD18 dimer, which forms a complex with neutrophils and induces intercellular adhesion molecule-1 (ICAM-1). The resulting tetrameric network (CD11b/CD18/neutrophil/ICAM-1) facilitates release of ROS after activation of neutrophil NADPH oxidase (NOX1), which are responsible for gastric epithelial cells apoptosis [54]. ROS also increase NF-kB expression by stimulating mitogen-activated protein kinases (MAPKs). These MAPKs are involved in the activation of AP-1 transcription factor, which will induce a pro-inflammatory cascade as well, often acting in conjunction with NF-kB [55]. The two transcription factors, NF-kB and Ap-1, own regulatory effects on IL-8 expression [56]. Therefore, IL-8 plays a central role in *H. pylori*-driven gastric carcinogenesis, by promoting angiogenesis, release of ROS and facilitating the persistence of a gastric “inflammatory environment”.

CagA and VacA, probably the most studied *H. pylori* virulence factors, also seem to be involved in this complex pathway to carcinogenesis. Type IV secretion system (T4SS) delivers CagA within gastric epithelial cells (Figure 1). Although CagA might not be essential for augmenting IL-8 expression [53], its role in oncogenesis has been attributed to promotion of epithelial–mesenchymal transition (EMT), strongly induced by downregulation of programmed cell death protein 4 (PDCD4) [57]. Contrariwise, VacA is a strong inducer of IL-8 expression and it also favors accumulation of CagA in gastric epithelial cells after disruption of autophagy mechanisms [45,58]. Thus, the most intensely studied *H. pylori* toxins, CagA and VacA, act synergistically and both lead to EMT, the main biological process responsible for invasion and metastasis of epithelial cells.

## 4. MiRNAs as Modulators of Innate Immune Response and *H. pylori* Associated Inflammation

Multiple studies have described miRNAs as modulators of TLR signaling, either acting as their direct ligands or as potential regulators of transcription. On the other hand, activation of TLRs, especially of those NF-kB-dependent can alter expression of miRNAs [59]. Inflammatory response can also be significantly influenced by these entities, as miRNAs modulate release of pro-inflammatory cytokines, transcription factors and cascade proteins induced by TLR signaling [60]. Therefore, TLR- miRNA crosstalk is essential for ensuring an adequate host immune response to pathogen. Given that *H. pylori* initiates an innate immune response induced by LPS–TLR interactions, miRNAs serve multiple regulatory functions in infection pathogenesis. The following subchapters highlight the involvement of several miRNAs, quantified in gastric tissue samples in the development of *H. pylori*-induced immune response and inflammation, based on current data. Figure 3 also illustrates involvement of these entities in modulation of NF-kB-dependent pathways.

### 4.1. Let-7 Family of miRNAs

MiRNA lethal-7 (let-7) was the first studied miRNA in humans. Its family encompasses so far more than 10 members. Let-7 activity is regulated by the LIN28A/LIN28B proteins, RNA binding proteins, which post-transcriptionally repress its biogenesis [61]. Apart from its multiple roles in carcinogenesis, let-7 is also involved in triggering immune responses, as let-7b binds to the 3′-UTR region of TLR4 mRNA and normally suppresses its activity at a post-transcriptional level [62,63]. *H. pylori* infection decreases let-7b levels, consequently upregulating TLR4 expression, and activating NF-κB, eventually leading to inflammation [63]. This finding is consistent with the above-mentioned study conducted by Isomoto et al., which reports a negative correlation between let-7b and IL-1b levels [64]. *H. pylori* CagA positive strains reduce expression of let-7 family members. Thus, it is unsurprising that let-7 expression has been negatively correlated with gastritis activity and severity scores, as provided by the updated Sydney classifications [47,65]. Let-7 family involvement in apoptosis and cell differentiation have also created the premises for further research regarding its relationship with gastric cancer, their expression being decreased in various cancers, including gastric carcinoma [66,67].

### 4.2. MiRNA-155

MiRNA-155, one of the most widely studied miRNAs, was also related to pathways of innate immunity [68,69]. Its expression is induced by activation of TLRs, through the MyD88- and Trif- dependent signaling pathways [70]. An alternative activation pathway of miRNA-155 is represented by T4SS, induced by PAMP receptors [71]. Furthermore, miRNA-155 expression is upregulated by release of TNF-α, commonly encountered during both bacterial and viral infections [70]. On the other hand, miRNA-155 also seems to be involved in a negative feedback mechanism, by attenuating NF-κB response and inhibiting the release of pro-inflammatory cytokines such as IL-8 in the setting of *H. pylori* infection [72,73]. Therefore, attenuation of inflammation favors bacterial persistence, as well as resistance to apoptosis within the macrophages [74]. MiRNA-155 has an essential contribution in promoting the activity of T cells, its expression inducing Th1 and Th17 differentiation, defined as instruments of adaptive immune response. This theory was sustained by a study conducted on mice, which concluded that subjects which were miRNA-155 deficient developed less severe *H. pylori*-associated gastric conditions, such as gastric atrophy, intestinal metaplasia or epithelial hyperplasia [75]. Moreover, another study conducted on a similar population concluded that lack of miRNA-155 expression will result in defective antitumor immunity [76].

### 4.3. MiRNA-146

MiRNA-146 is upregulated in similar manner to miRNA-155 as a response to bacterial pathogens, including *H. pylori* [77]. Activation of TLR2, TLR4, and TLR5 will result in activation of NF-κB, further inducing miRNA-146a/b expression [68]. MiRNA-146 targets tumor necrosis factor receptor associated factor-6 (TRAF6) and IL-1 receptor associated kinase 1 (IRAK1), which modulate TLR/ NF-κB pathway [78]. These two receptors have been used as signaling targets by multiple TLRs, including TLR2, TLR5, TLR7, TLR8, and TLR9, and the IL-1β receptor [79]. Considering that miRNA-146 is able to induce a negative feedback mechanism via downregulation of TRAF6 and IRAK1, inhibiting the release of pro-inflammatory cytokines, miRNA-146 can be considered as a modulator of multiple TLRs, with Nahid et al. describing it as a key effector in LPS-induced tolerance [80]. The same authors underlined in another study the role of miRNA-146a in conjunction with miRNA-132 and miRNA-212 as dampeners of an excessive immune response [81]. An additional role of miRNA-146 as a negative modulator of IL-8, growth-related oncogene (GRO)-α and macrophage inflammatory protein (MIP) -3α through the same NF-κB pathway was also reported as a result of *H. pylori* infection [82].

Considering the similar augmented expression of miRNA-155 and miRNA-146a as a result of *H. pylori* infection, Marquez et al. aimed to compare whether expression of these two miRNAs varied between pediatric, adult patients and an animal model in the setting of *H. pylori*-induced gastritis. An increased expression of both miRNAs was noticed in adult and pediatric patients with *H. pylori*-gastritis in comparison to subjects diagnosed with non-*H. pylori* gastritis, but significant differences were noted only in adults. Moreover, the increase in miRNA-155 and miRNA-146a expression was closely related to gastritis severity and infection chronicity, significantly higher differences in expression being suggestive of follicular gastritis and intestinal metaplasia in adult patients [78]. Similarly, upregulated expression of miRNA-155 was noted after 6 months of infection in mice, whereas increase in miRNA-146a was only identified after 18 months of persistent bacterial colonization. Still, expression level was significantly lower in mice than in humans, the authors attributing this finding to the shorter period of continuous exposure to the pathogen, without exceeding 18 months [78].

Thus, use of miRNA-155 and miRNA-146a as biomarkers of follicular gastritis and intestinal metaplasia could represent a gate towards early diagnosis. Considering the comparison of different infection models, we might state that duration of *H. pylori* infection correlates with the expression of these two miRNAs. Furthermore, this study is the first one yet to offer insights regarding the effects upon miRNA in pediatric *H. pylori*-induced gastritis.

### 4.4. MiRNA-125

Unlike miRNA-155, expression of miRNA-125b is reduced after exposure to LPS and/or bacterial pathogens. MiRNa-125b can target 3′-UTR of TNF-α transcripts, which suggests that its downregulation is mandatory to ensure that an inflammatory response will take place. Furthermore, LPS/TNF-α pathway is also involved in the genesis of endotoxin shock, suggesting that miRNA-125b may impact systemic response to bacterial pathogens [83]. A study performed in Brazil reported significant decrease in miRNA-125a-5p (common core sequence with miRNA-125b [68]) expression in the presence of *H. pylori* infection. *H. pylori* positive controls (without microscopic changes of the gastric mucosa), chronic gastritis, and gastric cancer patients exhibited significantly lower miRNA-125a-5p levels than *H. pylori* negative controls. The study also highlights lack of important expression variation, when comparing the three study groups as a whole, without taking into account *H. pylori* infection status. Therefore, the authors underlined the impact that the bacterial pathogen has upon miRNA-125a-5p regulation, as opposed to gastric inflammation or malignancy alone [84]. These findings also support an older theory suggesting that miRNA-125a-5p may represent a tumor suppressor, which inhibits invasive and metastatic features of malignant gastric cells by targeting breast cancer metastasis suppressor 1 (BRMS1) [85]. These results suggest that downregulated miRNA-125a-5p might be an accurate indicator of chronic gastritis and consequently of its potential evolution towards gastric cancer.

### 4.5. MiRNA-21

MiRNA-21 is a post-transcriptional suppressor of PDCD4, a protein that activates NF-κB and inhibits IL-10 production. Therefore, miRNA-21 enhances blockage of NF-κB and increases IL-10 levels, which limit inflammation [86,87]. The afore-mentioned protein, PCDCD4, is upregulated as a result of apoptotic processes [88]. Consequently, PCDCD4 acts as a tumor suppressor, whereas miRNA-21 is a promoter of carcinogenesis, whose overexpression was noted in several types of malignancies [89]. Upregulation of miRNA-21 has been encountered in gastric epithelial cells as a result of *H. pylori* infection and its persistence in the setting of gastric cancer suggests that this pathogen impairs the proliferation/apoptosis balance [90]. Contrariwise, a study comparing expression of various microRNAs between *H. pylori*-induced gastritis and gastric mucosa-associated lymphoid tissue (MALT) lymphoma revealed lack of important differences in miRNA-21 expression between the two groups [69]. In spite of these controversial findings, upregulated miRNA-21 could represent a potential marker for *H. pylori* infection, but further studies are required to define its precise role in bridging the progression of gastritis towards gastric malignancies.

### 4.6. MiRNA-223

MiRNA-223 expression is downregulated as a result of monocyte differentiation into macrophages. Its overexpression suppresses IL-1β release from the NLRP3 inflammasome [91]. Promoter region of miRNA-223 contains a binding site for NF-κB [92]. IRAK-1, MAPK, and NF-κB, inflammatory signals targeted by *H. pylori,* are negatively modulated by miRNA-223 in infected macrophages. Expression of inflammatory cytokines, TNF-α, IL-6, IL-8, and IL-12 is repressed as a result of these suppressed pathways [93]. Neutrophil infiltration may positively influence miRNA-223 expression, as eradication of *H. pylori* and complete disappearance of neutrophils from lamina propria normalizes its levels [65]. Thus, degree of miRNA-223 expression might be suggestive of gastritis activity. Moreover, an experimental study conducted on gastric cancer cells, infected with a CagA *H. pylori* strain revealed an increase in miRNA-223 expression through NF-κB signaling and also a downregulation of ARID1A (AT-rich interacting domain containing protein 1A), an inhibitor of carcinogenesis [94]. Taking into account that its upregulation is directly related with the amount of neutrophils within the gastric mucosa, which represent an indicator of inflammation activity, we can emphasize the role of miRNA-223 in predicting gastritis severity. Therefore, the overexpression of miRNA-223 might be considered a potential gastric cancer biomarker.

### 4.7. Other miRNAs and H. pylori

Release of pro-inflammatory cytokines plays a major role in the evolvement of histopathological changes within the gastric mucosa. Isomoto et al. conducted a study focusing on the expression of 29 miRNAs in relation to the release of four inflammatory cytokines, involved in the pathogenesis of *H. pylori*-associated gastritis: IL-1β, IL-6, IL-8, and TNF-α [64]. VacA plays an important role in their release, by activating mast cells [42]. An association between modified expression levels and at least one cytokine was found in 17 miRNAs. Five miRNAs were significantly correlated with the release of each of the four studied cytokines, in a negative manner: miRNA-103, miRNA-200b, miRNA-200c, miRNA-375, and miRNA-532. Upregulated expression in relation to activation of immune response was only found in three miRNAs, miRNA-204, miRNA-214, and miRNA-223. The study also reported a positive association between the four studied cytokines and chronic gastritis, as well as a significant predictive value of decreased levels of IL-6 and IL-8 in relation to intestinal metaplasia [64]. Most likely, the downregulation of the five aforementioned miRNAs defines their usefulness in detecting chronic gastric inflammation.

Another study analyzing the expression of multiple miRNAs in relation to *H. pylori* infection revealed significant differences in 55 of them. Still, only eight miRNAs qualified as possible discriminators of *H. pylori* infectious status: miRNA-203, miRNA-200a, miRNA-31, let-7e, miRNA-141, miRNA-203, miRNA-204, and miRNA-455. Out of these, miRNA-203 was the only one that underwent upregulation, the others being repressed in comparison to controls. Eradication of *H. pylori* enabled the return to normal levels of miRNAs belonging to the let-7 family, miRNA-200 family, miRNA-141, miRNA-130a, miRNA-106b, miRNA-31, miRNA-500, miRNA-532, and miRNA-652. CagA strains caused significant differences in expressions of miRNA-let-7a, miRNA-let-7d, miRNA-let-7f, miRNA-125a, and miRNA-500 when compared to wild strains. Three miRNAs (miRNA-223, miRNA-375, and miRNA-200c) were also significantly associated with gastritis activity scores, chronic inflammation and *H. pylori* colonization density scores [65].

The impact of *H. pylori* eradication schemes in terms of miRNAs expression have also been highlighted by a study conducted by Rossi et al., involving five miRNAs, namely miRNA-103a-3p, miRNA-181c-5p, miRNA-370-3p, miRNA-375, and miRNA-223-3p [95]. The authors pointed out that all five miRNAs were downregulated as a result of *H. pylori* infection, but the triple therapy regimen (with amoxicillin, clarithromycin and omeprazole) used for *H. pylori* eradication also induced the normalization of the expression in most of these miRNAs, miRNA-223-3p being the only exception, whose decreased expression persisted after treatment [95]. Therefore, besides their efficacy in terms of *H. pylori* eradication, therapeutic regimens might also improve the alterations in miRNAs-modulated signaling pathways, suggesting their subsequent role in hindering gastric carcinogenesis. Still, considering the limited data on this topic, further research is needed to establish how miRNAs are affected by *H. pylori* eradication.

Data on children are, however, limited. Except for the study comparing expression of miRNA-155 and miRNA-146a in an adult, pediatric, and animal model [78], only two other studies focused on miRNA and *H. pylori* infection in children, related to extragastric diseases [96,97]. According to a study conducted on a small population sample, miRNA-204 expression is decreased in children with pulpitis and *H. pylori*, as opposed to subjects with the same pathology of the oral cavity and negative *H. pylori* status [96]. Another small-scale, case-control study revealed a positive association between *H. pylori*-induced pediatric enteritis and upregulation of miRNA-32-5p [97]. These results render new research opportunities, not only in terms of *H. pylori*-associated alterations of miRNAs in children, but also regarding extragastric pathologies of the digestive tract caused by this bacterium.

Considering the above-mentioned findings, the research conducted on these miRNAs provides insights on their possible involvement in the genesis of acute, chronic, and pre-malignant lesions of the gastric mucosa caused by *H. pylori*, but also underlines the need for future research on this issue, involving other types of miRNAs and, especially, pediatric populations.

## 5. Conclusions

MiRNAs act as fine modulators of pathways involved in the pathogenesis of *H. pylori* infection. Therefore, they could be used as potential biomarkers, able to discriminate pathogen presence within the gastric mucosa, as well as its complications. Altered miRNA expression seems to be positively influenced by *H. pylori* eradication, but the information remains scarce on this topic. Nevertheless, data are limited regarding the precise mechanisms involved in miRNA expression changes. Research studies conducted so far relate expression alterations of several miRNAs to TLR–LPS interactions and activation of afferent pathways. Therefore, miRNAs modulate development of gastric inflammation and can influence its progression towards premalignant lesions or gastric cancer. Given the data regarding the precise implications are limited for multiple miRNAs, especially in children, future studies in this field are necessary to deliver answers about the exact involvement of these entities in the progression of *H. pylori*-associated gastropathy, involving multiple age groups.

## Figures and Tables

**Figure 1 ijms-22-01406-f001:**
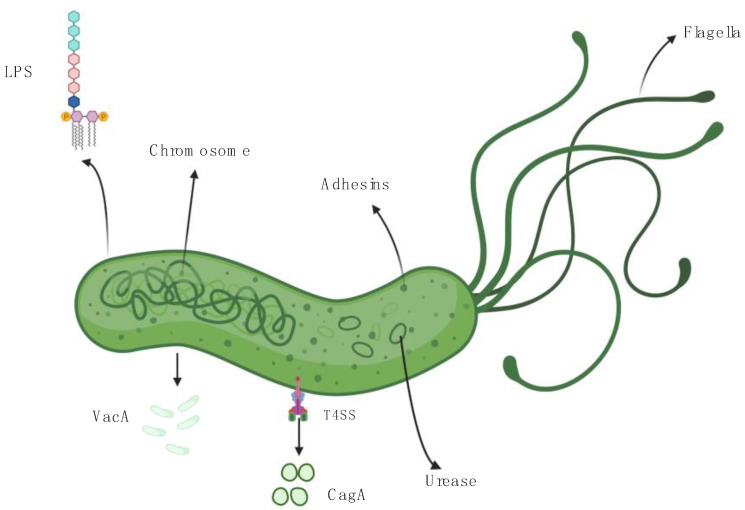
Insights of *H. pylori* structure and virulence factors. Created with BioRender.com (https://biorender.com/). *H. pylori* presents a unique helicoidal shape, which ensures effective colonization of the gastric mucosa. Flagella are essential for its mobility, whereas LPS is responsible for triggering host immune response. The urease, the main enzyme of the bacterium, is located inside the cytoplasm, where it maintains a constant alkaline pH, which facilitates efficient survival of the pathogen inside the gastric acid environment. VacA and CagA are two of *H. pylori*’s main virulence factors. Once delivered to host cells, these two entities can mediate inflammatory pathways leading to carcinogenesis. Legend: CagA—cytotoxin-associated gene A; LPS—lipopolysaccharide; T4SS—type IV secretion system; VacA—vacuolating cytotoxin.

**Figure 2 ijms-22-01406-f002:**
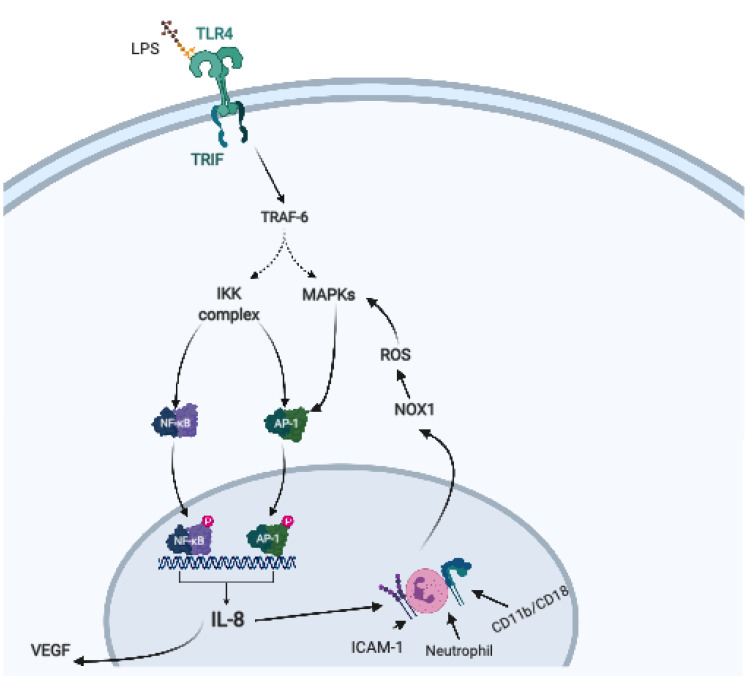
The relationship between IL-8 and gastric carcinogenesis. Created with BioRender.com (https://biorender.com/) IL-8 expression is upregulated as a result of *H. pylori* infection and plays a central role in the development of gastric cancer. This cytokine is released after activation of NF-kB- and AP-1- dependent pathways. Its chemotactic properties for neutrophils will eventually facilitate formation of a tetrameric network (CD11b/CD18/neutrophil/ICAM-1), which will induce ROS release through activation of NOX-1. ROS damage epithelial cells and stimulate MAPKs, which further enhance AP-a transcription and, subsequently, IL-8 release. The cytokine is also responsible for VEGF release, with angiogenic properties. Legend: AP-1—activator protein 1; ICAM-1—intercellular adhesion molecule-1; IKK—IκB kinase; IL—interleukin; LPS—lipopolysaccharide; MAPK—mitogen-activated protein kinase; NF-kB—nuclear factor kappa B; NOX-1—neutrophil NADPH oxidase; ROS—reactive oxygen species; TLR—toll-like receptor; TRAF6—tumor necrosis factor receptor associated factor-6; VEGF—vascular endothelial growth factor.

**Figure 3 ijms-22-01406-f003:**
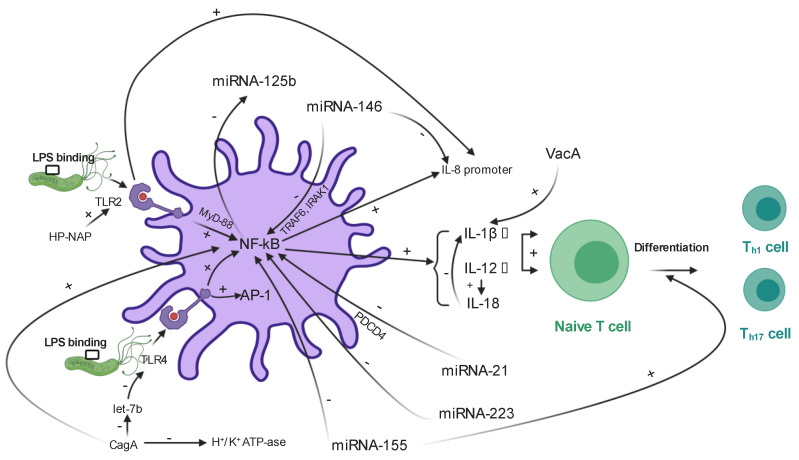
MiRNA involvement in *H. pylori* nuclear factor-kappa B (NF-kB) dependent pathways. Created with BioRender.com (https://biorender.com/). NF-kB dependent pathways are activated as a result of TLR-2/TLR-4 and LPS interactions and stimulate release of pro-inflammatory cytokines, including IL-12 and IL-1β, responsible for differentiation of T Helper 1 and Th17 cells. The three main virulence factors of *H. pylori* positively impact release of inflammatory cytokines, VacA being the only one, which is not directly involved in TLR–LPS interaction. Illustrated miRNAs seem to be able to both negatively modulate NF-kB expression and release of inflammatory cytokines. Furthermore, a few of them are downregulated by CagA (i.e., let-7b) and by NF-kB (i.e., miRNA-155). Legend: CagA—cytotoxin-associated gene A; HP-NAP—*H. pylori* neutrophil-activating protein; IL—interleukin; Let-7—lethal 7; LPS—lipopolysaccharide; miRNA—microRNA; NF-kB—nuclear factor kappa B; Th—T helper cell; TLR—toll-like receptor; VacA—vacuolating cytotoxin.

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
