# Peer review of "MicroRNA Modulation of Host Immune Response and Inflammation Triggered by Helicobacter pylori"

_ijms, 2021, doi:10.3390/ijms22031406_

Round 1

Reviewer 1 Report

In this review, Săsăran et al highlight the role of miRNAs provoked by H. pylori infection. The manuscript needs to be checked for grammar and spelling mistakes and the sentences need to be written more clearly. In addition, some figures would be a great addition. I have listed my remarks and recommendations:

General remarks and recommendations

  • The manuscript needs a thorough spelling and grammar check. A lot of sentences are too complicated or wrong which makes it hard to read, understand and review. Some examples are
    • Premises (line 85) or premisses (line 72) are both used
    • Line 177: Expectable = expected
  • Please make at least 2 figures: one representing the h pylori structure and one about the immune response induced by h pylori and where the miR interfere
  • I would recommend the authors to focus more on the use of these miR as potential bio-markers than on their therapeutic potential
  • Certain observations are only briefly mentioned but should be discussed more in dept
  • Can the authors add a short paragraph that describes the course of a H pylori infection and how this can lead to cancer. If not to complicated, signaling pathways can also be mentioned
  • Unfortunately, in addition to the spelling and grammar mistakes, a lot of the sentences written in the manuscript are very unclear. The author need to rephrase the sentences or clarify what they mean. Some (but not the only) examples
    • Line 105: HupA, its main enzyme, favours antibiotic resistance [26].
    • Lines 42-44: H. pylori presents a lipopolysaccharide (LPS), virulence factor and even the capacity of forming a biofilm, which helps in neutralizing immune response [6,7]
    • Line 116 molecular peculiarities of the pathogen
    • Line 153 being able to transcript oncogenic genes
  • I would suggest to first use the part on miRs in general and then discuss some specific miRs in the next parts.
    • 7].

Specific comments

  • Lines 47-49: Still, treatment of H. pylori can also positively impact precancerous lesions, by improving glandular atrophy, as well as intestinal metaplasia, in a more selected number of cases [9,10].
    à what type of treatment?
  • Line 163-166: Infection persistence will determine arousal of lymphocytes within the gastric mucosa, which represent a sign of chronic inflammation. Therefore, lymphocytes are more often identified in adult gastric biopsy specimens than in those from children, concomitant presence of neutrophils suggesting an active, chronic gastritis [46].
    à Elaborate please, does this mean that throughout childhood this persist? Do the adults have history of pylori infection when they were young
  • Line 169-171: Reproduction of an infection model during an in-vitro study, conducted on mice, revealed that an infiltration of macrophages and neutrophils occurs immediately after inoculation of H. pylori, in the first two days. However, their number subsides gradually and increases again 2-3 weeks post-infection
    à What is infiltrated? In vitro study conducted on mice?
  • Lines 206-27: Let-7 activity is regulated by the LIN28A/LIN28B proteins, RNA binding proteins, which post-transcriptionally repress its function [54]
    à What function?
  • Lines 234-238: miRNA-155 deficient developed less severe H. pylori associated gastric conditions, such as gastric atrophy, intestinal metaplasia or epithelial hyperplasia [68]. On the other hand, another study conducted on a similar population concluded that lack of miRNA-155 expression will result in defective antitumor immunity. The authors proposed upregulation of miRNA-155 as a potential novel therapeutic approach in cancers [69].
    à Discuss more in dept especially why a better phenotype in miRNA deficient mice leads to a proposal to upregulate miRNA155 as potential novel therapeutics
  • Lines 256-258: Considering the strong resemblances that H. pylori infections poses upon miRNA-155 and miRNA-146a expression after TLR activation, Marquez et al. aimed to compare whether expression of these two miRNAs varied between paediatric, adult patients and an animal model in the setting of H. pylori- induced gastritis
    à clarify the first part of this sentences and why are the results of the animal model not discussed
  • Lines 276-2778: The study also highlights lack of important expression variation, when comparing the study groups as a whole, without taking into account H. pylori infection status. Therefore, this article underlines the impact that the bacterial pathogen has upon miRNA 125a-5p regulation, as opposed to gastric inflammation or malignancy alone [77]
    à explain
  • Lines 289-309 Upregulation of miRNA-21 has been reported in gastric epithelial cells as a result of H. pylori infection and persists in gastric cancers, suggesting that persistence of this pathogen is the main disruptor of proliferation/apoptosis balance [83]
    à explain
  • Lines 306-308: Moreover, CagA H. pylori strains specifically increase miRNA-223 expression through NF-κB signaling and also downregulate ARID1A (AT-rich interacting domain containing protein 1A), an inhibitor of carcinogenesis [87]
    à Clarify please, is this only for CagA H pylori strains or also others
  • Lines 355-356: Therefore, they could be used as potential non-invasive biomarkers, capable of discriminating the presence of pathogen within the gastric mucosa, as well as its complications
    à is there more info on the expression of these miR throughout the course of infection? please address this

Author Response

January the 25th, 2020

To the Editorial Board of International Journal of Molecular Sciences,

Dear Editor,

Please find attached a revised version of the manuscript entitled: " International Journal of Molecular Sciences," written by Maria Oana Săsăran, Lorena Elena Meliţ and Ecaterina Daniela Dobru, Manuscript ijms-1090559

Firstly, we thank very much the reviewers for their valuable comments and suggestions in order to improve our paper.

Following the reviewers’ concerns and observations, we made some modifications to the initial version of our manuscript, which we described in detail, according to their recommendations, highlighting them in yellow in the attached manuscript. The number of the lines mentioned below can be found in the  “Manuscript SMO Changes accepted_MicroARN_IJMS_ 25.01.2021” uploaded document.

Reviewer 1

Open Review

(x) I would not like to sign my review report

( ) I would like to sign my review report

English language and style

(x) Extensive editing of English language and style required

( ) Moderate English changes required

() English language and style are fine/minor spell check required

( ) I don't feel qualified to judge about the English language and style

Comments and Suggestions for Authors

In this review, Săsăran et al highlight the role of miRNAs provoked by H. pylori infection. The manuscript needs to be checked for grammar and spelling mistakes and the sentences need to be written more clearly. In addition, some figures would be a great addition. I have listed my remarks and recommendations:

General remarks and recommendations

Comment 1 The manuscript needs a thorough spelling and grammar check. A lot of sentences are too complicated or wrong which makes it hard to read, understand and review. Some examples are

    • Premises (line 85) or premisses (line 72) are both used
    • Line 177: Expectable = expected

Answer 1

Please apologize for our mistakes. The manuscript was revised regarding spelling and grammar corrections by a native English speaker and the unclear statements were rephrased taking into account your specific comments.

Comment 2 Please make at least 2 figures: one representing the h pylori structure and one about the immune response induced by h pylori and where the miR interfere

Thank you for your extremely useful suggestion. We have designed three figures using the Biorender app, one regarding H. pylori structure, a second one with activation of NF-kB- dependent pathway and miRNA interference, and a third one with IL-8 dependent carcinogenic pathways. These three figures have also been citated inside the text:

Lines 738: “Figure 1 – Insights of H. pylori structure and virulence factors.”

Lines 743: “Figure 2 – The relationship between IL-8 and gastric carcinogenesis.”

Lines 750:”Figure 3 – MiRNAs involvement in H. pylori NF-kB-dependent pathways.”

Comment 3. I would recommend the authors to focus more on the use of these miR as potential bio-markers than on their therapeutic potential

Answer 3

Thank you for your suggestions. We have added concluding remarks where necessary in the paragraphs focusing upon individual miRNAs, in order to underline their potential role as bio-markers. Therefore, we added the following paragraphs in the revised form of our manuscript:

Lines 320-323: “Thus, use of miRNA-155 and miRNA-146a as biomarkers of follicular gastritis and intestinal metaplasia could represent a gate towards early diagnosis. Considering the comparison of different infection models, we might state that duration of H. pylori infection correlates with the expression of these two miRNAs.”

Lines 344-346: “These results suggest that downregulated miRNA-125a-5p might be an accurate indicator of chronic gastritis and consequently of its potential evolution towards gastric cancer.”

Lines 360-363: “In spite of these controversial findings, upregulated miRNA-21 could represent a potential marker for H. pylori infection, but further studies are required to define its precise role in bridging the progression of gastritis towards gastric malignancies.”

Lines 378-382: “Taking into account that its upregulation is directly related with the amount of neutrophils within the gastric mucosa, which represent an indicator of inflammation activity, we can emphasize the role of miRNA-223 in predicting gastritis severity. Therefore, the overexpression of miRNA-223 might be considered a potential gastric cancer biomarker.”

Lines 397-398: “Most likely, the downregulation of the five aforementioned miRNAs defines their usefulness in detecting chronic gastric inflammation.”

Comment 4. Certain observations are only briefly mentioned but should be discussed more in dept

Answer 4

We greatly appreciate your suggestion. We have tried to rephrase unclear sentences and detail those which were too brief, also taking into account your specific comments. Furthermore, we hope that the new provided illustrations will also be helpful for a better understanding of signaling pathways and H. pylori structure.

Comment 5. Can the authors add a short paragraph that describes the course of a H pylori infection and how this can lead to cancer. If not to complicated, signaling pathways can also be mentioned

Answer 5

Thank you for your recommendation. We have added a separate shorter subchapter, named “Helicobacter pylori-driven pathways of gastric carcinogenesis” synthesizing the signaling pathways leading to gastric cancer associated with innate immune response, focusing on the role of IL-8 and the involvement of CagA and VacA, two well-documented H. pylori virulence factors. Furthermore, we also introduced a figure illustrating these pathways (figure 2).

Lines 204-236: “Involvement of H. pylori in the development of gastric cancer represents an area of continuous research, with still incompletely described signaling pathways. Innate immune response seems to trigger multiple key effectors of gastric carcinogenesis, whereas virulence factors also play their roles in this process.

As previously described, activation of NF-kB-dependent pathways leads to release of various inflammatory cytokines. MD-2 protein facilitates recognition of LPS by TLR-4, interaction of these two entities resulting not only in the activation of NF-kB, but also in augmentation of IL-8 promoter function [52]. IL-8 can significantly contribute to gastric oncogenesis, as represented in figure 2. IL-8 acts as a chemotactic agent for neutrophils, which corelate with gastritis severity and stimulate release of vascular endothelial growth factor (VEGF), as well as proliferation and migration of endothelial cells, subsequently promoting angiogenesis [53]. This chemotaxis is achieved via CD11b/CD18 dimer, which forms a complex with neutrophils and induces intercellular adhesion molecule-1 (ICAM-1). The resulting tetrameric network (CD11b/CD18/neutrophil/ICAM-1) facilitates release of ROS after activation of neutrophil NADPH oxidase (NOX1), which are responsible for gastric epithelial cells apoptosis [54]. ROS also increase NF-kB expression by stimulating mitogen-activated protein kinases (MAPKs). These MAPKs are involved in the activation of AP-1 transcription factor which will induce a pro-inflammatory cascade as well, often acting in conjunction with NF-kB [55]. The two transcription factors, NF-kB and Ap-1, own regulatory effects on IL-8 expression [56]. Therefore, IL-8 plays a central role in H. pylori- driven gastric carcinogenesis, by promoting angiogenesis, release of ROS and facilitating the persistence of a gastric ‘inflammatory environment’.

CagA and VacA, probably the most studied H. pylori virulence factors, also seem to be involved in this complex pathway to carcinogenesis. Type IV secretion system (T4SS) delivers CagA within gastric epithelial cells (figure 1). Although CagA might not be essential for augmenting IL-8 expression [53], its role in oncogenesis has been attributed to promotion of epithelial-mesenchymal transition (EMT), strongly induced by downregulation of programmed cell death protein 4 (PDCD4) [57]. Contrariwise, VacA is a strong inducer of IL-8 expression and it also favors accumulation of CagA in gastric epithelial cells after disruption of autophagy mechanisms [45,58]. Thus, the most intensely studied H. pylori toxins, CagA and VacA, act synergistically and both lead to EMT, the main biological process responsible for invasion and metastasis of epithelial cells.”

Comment 6.

Unfortunately, in addition to the spelling and grammar mistakes, a lot of the sentences written in the manuscript are very unclear. The author need to rephrase the sentences or clarify what they mean. Some (but not the only) examples

    • Line 105: HupA, its main enzyme, favours antibiotic resistance [26].
    • Lines 42-44: H. pylori presents a lipopolysaccharide (LPS), virulence factor and even the capacity of forming a biofilm, which helps in neutralizing immune response [6,7]
    • Line 116 molecular peculiarities of the pathogen
    • Line 153 being able to transcript oncogenic genes

Answer 6

Thank you for your suggestion. The phrases you mentioned above along with other we considered unclear underwent significant modifications and/or accompanying explanations, in order to provide clearer information:

Lines 116-119: “HupA, the main enzyme of LPS, favors resistance to cationic antimicrobial peptides (CAMPs) through a currently unknown process, presumptively inducing changes in phospholipid synthesis. Survival of H. pylori is thus ensured, as well its capacity to evade innate immune response.”

Lines 45-48: “Biofilm formation has been attributed to morphological changes (cell wall rearrangement), membrane vesicles secretion and inter-microbial communication, which constitute virulence and adaptive responses of the pathogen [6,7].”

Lines 129-130: “Alteration of the epithelial barrier will set off innate detection mechanism, implying recognition of pathogen specific molecular structures.”

Line 164: “being able to transcript oncogenes”

Comment 7. I would suggest to first use the part on miRs in general and then discuss some specific miRs in the next parts.

Thank you for your recommendation. Data regarding the role of miRNAs in general is included in the introduction section. New data regarding their cellular source and potential extracellular identification situses has been added according to reviewer’s 2 recommendations (lines 76-82): “MiRNAs have a cellular source, their biogenesis implying multiple nuclear and cytoplasmatic maturation processes being eventually released in the extracellular compartment. Their extracellular delivery is ensured by exosomes, microvesicles, protein and lipoprotein complexes. MiRNAs have been identified in numerous extracellular bio-fluids, including serum, plasma, urine, saliva, gastric juice, bronchial lavage, pleural peritoneal or synovial fluids [20]. Therefore, quantifying their expression does not always imply the assessment of tissue biopsies, being also achieved using various bio-fluid samples.”

We have removed data regarding miRNA-146a from the introduction section. Furthermore, an introductory paragraph has already been provided in the “MiRNAs as modulators of innate immune response and H. pylori – associated inflammation” chapter.

Specific comments 

Comment 1

Lines 47-49: Still, treatment of H. pylori can also positively impact precancerous lesions, by improving glandular atrophy, as well as intestinal metaplasia, in a more selected number of cases [9,10].
à what type of treatment?

Answer 1

Please accept our apologies for not mentioning clearly the type of treatment. When mentioning ‘treatment of H. pylori’, we wanted to underline the role of standard eradication therapeutic schemes used for H. pylori treatment. We have rephrased the sentence, clarifying the type of treatment (lines 51-53): “Nevertheless, standard eradication regimens recommended by current guidelines in treating H. pylori can also positively impact precancerous lesions, by improving glandular atrophy, as well as intestinal metaplasia, in a selected number of cases [9,10].”

Comment 2

Line 163-166: Infection persistence will determine arousal of lymphocytes within the gastric mucosa, which represent a sign of chronic inflammation. Therefore, lymphocytes are more often identified in adult gastric biopsy specimens than in those from children, concomitant presence of neutrophils suggesting an active, chronic gastritis [46].
à Elaborate please, does this mean that throughout childhood this persist? Do the adults have history of pylori infection when they were young

Answer 2

Thank you for your comment. It is well-known that H. pylori infection is usually acquired during childhood and persists into adulthood. We have inserted another phrase to emphasize this issue (lines 176-179):  “Taking into account that most individuals acquire the infection during childhood [1], these differences in histological findings between adults and children further sustain that its persistence over time results in chronic changes of the gastric mucosa.”

Comment 3

Line 169-171: Reproduction of an infection model during an in-vitro study, conducted on mice, revealed that an infiltration of macrophages and neutrophils occurs immediately after inoculation of H. pylori, in the first two days. However, their number subsides gradually and increases again 2-3 weeks post-infection
à What is infiltrated? In vitro study conducted on mice?

Answer 3

We apologize for our mistake and the confusion we created. We have removed the “in vitro” syntagm and clarified the infiltration of gastric mucosa with macrophages and neutrophils (lines 181-182): “Reproduction of an infection model on mice revealed that macrophage and neutrophil infiltration of the gastric mucosa occurs within two days after H. pylori inoculation.”

Comment 4

Lines 206-27: Let-7 activity is regulated by the LIN28A/LIN28B proteins, RNA binding proteins, which post-transcriptionally repress its function [54]
à What function?

Answer 4

Thank you for identifying this expression error. These proteins are involved in inhibiting biogenesis of let-7 family, as mentioned in line 256-257: “Let-7 activity is regulated by the LIN28A/LIN28B proteins, RNA binding proteins, which post-transcriptionally repress its biogenesis.”

Comment 5

Lines 234-238: miRNA-155 deficient developed less severe H. pylori associated gastric conditions, such as gastric atrophy, intestinal metaplasia or epithelial hyperplasia [68]. On the other hand, another study conducted on a similar population concluded that lack of miRNA-155 expression will result in defective antitumor immunity. The authors proposed upregulation of miRNA-155 as a potential novel therapeutic approach in cancers [69].
à Discuss more in dept especially why a better phenotype in miRNA deficient mice leads to a proposal to upregulate miRNA155 as potential novel therapeutics

Answer 5

Thank you for your suggestion. We have followed your recommendations and decided to focus more on miRNAs as potential biomarkers than on their potential role as therapeutic targets, given the insufficient data so far available. Therefore, we have removed this last sentence from the paragraph, related to potential novel therapeutic approaches.

Comment 6

Lines 256-258: Considering the strong resemblances that H. pylori infections poses upon miRNA-155 and miRNA-146a expression after TLR activation, Marquez et al. aimed to compare whether expression of these two miRNAs varied between paediatric, adult patients and an animal model in the setting of H. pylori- induced gastritis

Answer 6

The first part of the sentence was clarified. We were referring to the initial part of the paragraph describing how both miRNAs are upregulated in similar manner, after TLR signaling (lines 306-309): “Considering the similar augmented expression of miRNA-155 and miRNA-146a as a result of H. pylori infection, Marquez et al. aimed to compare whether expression of these two miRNAs varied between pediatric, adult patients and an animal model in the setting of H. pylori- induced gastritis.”

Moreover, we described the evolution of miRNA-155 and miRNA-146a expression after reproduction of the infection on mice and provided some further explanatory conclusions (lines 314-325): Similarly, upregulated expression of miRNA-155 was noted after 6 months of infection in mice, whereas increase in miRNA-146a was only identified after 18 months of persistent bacterial colonization. Still, expression level was significantly lower in mice than in humans, the authors attributing this finding to the shorter period of continuous exposure to the pathogen, without exceeding 18 months [78].

Thus, use of miRNA-155 and miRNA-146a as biomarkers of follicular gastritis and intestinal metaplasia could represent a gate towards early diagnosis. Considering the comparison of different infection models, we might state that duration of H. pylori infection correlates with the expression of these two miRNAs. Furthermore, this study is the first one yet to offer insights regarding the effects upon miRNA in pediatric H. pylori- induced gastritis.”

Comment 7

Lines 276-2778: The study also highlights lack of important expression variation, when comparing the study groups as a whole, without taking into account H. pylori infection status. Therefore, this article underlines the impact that the bacterial pathogen has upon miRNA 125a-5p regulation, as opposed to gastric inflammation or malignancy alone [77]
à explain

Answer 7

According to your recommendation we described more thoroughly the study mentioned above and we hope the information is now easier to understand. The paragraph has been rephrased and new lines have been added (lines 333-341): “A study performed in Brazil reported significant decrease in miRNA-125a-5p (common core sequence with miRNA-125b [68]) expression in the presence of H. pylori infection. H. pylori positive controls (without microscopic changes of the gastric mucosa), chronic gastritis and gastric cancer patients exhibited significantly lower miRNA-125a-5p levels than H. pylori negative controls. The study also highlights lack of important expression variation, when comparing the three study groups as a whole, without taking into account H. pylori infection status. Therefore, the authors underlined the impact that the bacterial pathogen has upon miRNA-125a-5p regulation, as opposed to gastric inflammation or malignancy alone [84].” 

Comment 8

Lines 289-309 Upregulation of miRNA-21 has been reported in gastric epithelial cells as a result of H. pylori infection and persists in gastric cancers, suggesting that persistence of this pathogen is the main disruptor of proliferation/apoptosis balance [83]
à explain

Answer 8

According to your recommendation, we have rephrased the above mentioned sentence in order to make it easier to understand (lines 354-357): “Upregulation of miRNA-21 has been encountered in gastric epithelial cells as a result of H. pylori infection and its persistence in the setting of gastric cancer suggests that this pathogen impairs the proliferation/apoptosis balance [90].”

Comment 9

Lines 306-308: Moreover, CagA H. pylori strains specifically increase miRNA-223 expression through NF-κB signaling and also downregulate ARID1A (AT-rich interacting domain containing protein 1A), an inhibitor of carcinogenesis [87]
à Clarify please, is this only for CagA H pylori strains or also others

Answer 9

This cited experimental study has only included gastric cancer cells infected with a CagA H. pylori strain. VacA seems to not be specifically involved in NF-κB signaling, but rather in modulation of pro-inflammatory cytokine levels, as illustrated in figure 3. We have rephrased this sentence, underlining how the study was conducted (lines 374-378): “Moreover, an experimental study conducted on gastric cancer cells, infected with a CagA H. pylori strain, revealed an increase in miRNA-223 expression through NF-κB signaling and also a downregulation of ARID1A (AT-rich interacting domain containing protein 1A), an inhibitor of carcinogenesis [94].”

Comment 10

Lines 355-356: Therefore, they could be used as potential non-invasive biomarkers, capable of discriminating the presence of pathogen within the gastric mucosa, as well as its complications
à is there more info on the expression of these miR throughout the course of infection? please address this

Answer 10

Eradication of H. pylori seems to positively impact altered miRNA expression. We have identified only two studies on this matter (references no. 65- already cited and no. 95). A new paragraph has been added, describing the study conducted by Rossi et al (reference no.96), lines 411-422: “The impact of H. pylori eradication schemes in terms of miRNAs expression have also been highlighted by a study conducted by Rossi et al., involving five miRNAs, namely miRNA‐103a‐3p, miRNA‐181c‐5p, miRNA‐370‐3p, miRNA‐375 and miRNA‐223‐3p [96]. The authors pointed out that all five miRNAs were downregulated as a result of H. pylori infection, but the triple therapy regimen (with amoxicillin, clarithromycin and omeprazole) used for H. pylori eradication also induced the normalization of the expression in most of these miRNAs, miRNA-223-3p being the only exception, whose decreased expression persisted after treatment [96]. Therefore, besides their efficacy in terms of H. pylori eradication, therapeutic regimens might also improve the alterations in miRNAs-modulated signaling pathways, suggesting their subsequent role in hindering gastric carcinogenesis. Still, considering the limited data on this topic, further research is needed to establish how miRNAs are affected by H. pylori eradication.”

The need for future research on this matter has also been highlighted in the Conclusions section (lines 443-445): “Altered miRNA expression seems to be positively influenced by H. pylori eradication, but the information remains scarce on this topic.”

Thus, by this letter and by the attached revised manuscript of our original manuscript, we hope to have fulfilled all the observations and recommendations made by the Reviewers.

Thank you for your time and consideration.

On behalf of all authors of this work,

Yours sincerely,

Lecturer Lorena Elena Meliț, MD, PhD

Departament of Pediatrics I, “George Emil Palade ”University of Medicine, Pharmacy, Sciences and Technology Târgu Mures, 38 Gh. Marinescu St., 540139, Târgu Mures, Romania. Phone: +40-742-984744. Fax: +40-265-211098, e-mail: lory_chimista89@yahoo.com

Reviewer 2 Report

Interesting review.

Some important aspects need attention:

  1. What is the cellular source of miRNAs-are the from gastric/duodenal cells, neutrophils, macrophages or any other source.
  2. If one want to look at them can they be found in the gastric juice/duodenal secretions and quantitate them.
  3. Are miRNAS be detected in the urine of the patients.
  4. Can MiRNAs be acted upon by various NSAIDs, H2 blockers, PPI, and antibiotics and if so what is their significance.  
  5. Are there any compounds that specifically suppress/increase the expression of miRNAs and if so their role in the treatment of DU/gastric ulcer and gastric cancer.
  6. How can this knowledge be used in the clinic.   

Author Response

January the 25th, 2020

To the Editorial Board of International Journal of Molecular Sciences,

Dear Editor,

Please find attached a revised version of the manuscript entitled: " International Journal of Molecular Sciences," written by Maria Oana Săsăran, Lorena Elena Meliţ and Ecaterina Daniela Dobru, Manuscript ijms-1090559

Firstly, we thank very much the reviewers for their valuable comments and suggestions in order to improve our paper.

Following the reviewers’ concerns and observations, we made some modifications to the initial version of our manuscript, which we described in detail, according to their recommendations, highlighting them in yellow in the attached manuscript. The number of the lines mentioned below can be found in the  “Manuscript SMO Changes accepted_MicroARN_IJMS_ 25.01.2021” uploaded document.

Reviewer 2

Open Review

( ) I would not like to sign my review report

(x) I would like to sign my review report

English language and style

( ) Extensive editing of English language and style required

() Moderate English changes required

(x) English language and style are fine/minor spell check required

( ) I don't feel qualified to judge about the English language and style

Comments and Suggestions for Authors

Comment 1

Interesting review.

Answer 1

Thank you very much for your positive comments.

Comment 2

What is the cellular source of miRNAs-are the from gastric/duodenal cells, neutrophils, macrophages or any other source.

Answer 2

We apologize for not mentioning this important detail initially. We have added a paragraph in the introduction section, briefly describing the cellular source of miRNAs, as well as their extracellular release within bio-fluids (lines 76-82): “MiRNAs have a cellular source, their biogenesis implying multiple nuclear and cytoplasmatic maturation processes being eventually released in the extracellular compartment. Their extracellular delivery is ensured by exosomes, microvesicles, protein and lipoprotein complexes. MiRNAs have been identified in numerous extracellular bio-fluids, including serum, plasma, urine, saliva, gastric juice, bronchial lavage, pleural peritoneal or synovial fluids [20]. Therefore, quantifying their expression does not always imply the assessment of tissue biopsies, being also achieved using various bio-fluid samples.”

Furthermore, we have mentioned that the studies reviewed in the miRNA specific section involved cellular miRNAs, depicted from the gastric mucosa (lines 247-251): “The following subchapters highlight the involvement of several miRNAs, quantified in gastric tissue samples, in the development of H. pylori- induced immune response and inflammation, based on current data.”

Comment 3

If one want to look at them can they be found in the gastric juice/duodenal secretions and quantitate them.

Answer 3

Most of the studies regarding H. pylori- associated pathologies quantified miRNAs using gastric biopsy samples. Nevertheless, it seems that they can also be identified in gastric juice. Here is an example of a study assessing miRNAs identified in the gastric juice (Tanaka et al. Exosomal hsa-miR-933 in Gastric Juice as a Potential Biomarker for Functional Dyspepsia. Dig Dis Sci. 2020 Dec;65(12):3493-3501. doi: 10.1007/s10620-020-06096-7). Our study mainly focused on miRNAs assessed in gastric sample biopsies, as we already mention in the first paragraph of the “MiRNAs as modulators of innate immune response and H. pylori – associated inflammation” chapter (lines 247-250): “The following subchapters highlight the involvement of several miRNAs, quantified in gastric tissue samples, in the development of H. pylori- induced immune response and inflammation, based on current data.”

Comment 4

Are miRNAS be detected in the urine of the patients.

Answer 4

MiRNAs can indeed be detected in urine, as well as in other bio-fluids, as highlighted by the paragraph added in the introduction section (already mentioned as a response to your initial comment), lines 79-81: “MiRNAs have been identified in numerous extracellular bio-fluids, including serum, plasma, urine, saliva, gastric juice, bronchial lavage, pleural peritoneal or synovial fluids [20].”

Comment 5

Can MiRNAs be acted upon by various NSAIDs, H2 blockers, PPI, and antibiotics and if so what is their significance.  

Answer 5

Unfortunately, only a few studies assessed this topic. Thus, two of them underlined how standard therapeutic eradication schemes of H. pylori can positively impact altered miRNA expression. One of them has already been cited (Matsushima et al.), while the findings of the other one (Rossi et al.) have now been added in the “Other miRNAs and H. pylori” subchapter. MiRNA expression seems to return back to normal after successful H. pylori eradication, but future studies should be able to provide answers on this matter. The following paragraph has been added (lines 411-422): “The impact of H. pylori eradication schemes in terms of miRNAs expression have also been highlighted by a study conducted by Rossi et al., involving five miRNAs, namely miRNA‐103a‐3p, miRNA‐181c‐5p, miRNA‐370‐3p, miRNA‐375 and miRNA‐223‐3p [95]. The authors pointed out that all five miRNAs were downregulated as a result of H. pylori infection, but the triple therapy regimen (with amoxicillin, clarithromycin and omeprazole) used for H. pylori eradication also induced the normalization of the expression in most of these miRNAs, miRNA-223-3p being the only exception, whose decreased expression persisted after treatment [95]. Therefore, besides their efficacy in terms of H. pylori eradication, therapeutic regimens might also improve the alterations in miRNAs-modulated signaling pathways, suggesting their subsequent role in hindering gastric carcinogenesis. Still, considering the limited data on this topic, further research is needed to establish how miRNAs are affected by H. pylori eradication.”

MiRNAs can be influenced by NSAIDs, also in relation to modulation of epigenetic mechanisms leading to carcinogenesis (Yiannakopoulou. Targeting epigenetic mechanisms and microRNAs by aspirin and other non steroidal anti-inflammatory agents--implications for cancer treatment and chemoprevention. Cell Oncol (Dordr) 2014 Jun;37(3):167-78. doi: 10.1007/s13402-014-0175-7). Still, there is no available data regarding potential role of NSAIDs in modulating expression of miRNAs involved in H. pylori or gastric cancer signaling pathways. Therefore, we did not consider that introduction of this data was appropriate in our review article. Furthermore, there is no data about implications of H2 blockers in association with miRNA expression.

Comment 6

Are there any compounds that specifically suppress/increase the expression of miRNAs and if so their role in the treatment of DU/gastric ulcer and gastric cancer.

Answer 6

Modulation of miRNA-dependent pathways, as well as of their promoters/inhibitors represents an area of ongoing research, without clearly established therapeutic solutions. Most studies available so far have only described miRNAs as potential biomarkers of H. pylori infection, gastritis or gastric cancer. Taking into account that the exact implications of multiple miRNAs is not yet known and in light of reviewer’s 1 comments, we have decided to remove conclusions regarding potential therapeutic possibilities involving miRNA up- or downregulation and to focus on their role as biomarkers.

Comment 7

How can this knowledge be used in the clinic. 

Answer 7

Thank you for your suggestion. We have added a concluding remark after discussing each individual miRNA, underlining their potential role as biomarkers of H. pylori- related gastric conditions. The following lines have been added:

Lines 320-323: “Thus, use of miRNA-155 and miRNA-146a as biomarkers of follicular gastritis and intestinal metaplasia could represent a gate towards early diagnosis. Considering the comparison of different infection models, we might state that duration of H. pylori infection correlates with the expression of these two miRNAs.”

Lines 344-346: “These results suggest that downregulated miRNA-125a-5p might be an accurate indicator of chronic gastritis and consequently of its potential evolution towards gastric cancer.”

Lines 360-363: “In spite of these controversial findings, upregulated miRNA-21 could represent a potential marker for H. pylori infection, but further studies are required to define its precise role in bridging the progression of gastritis towards gastric malignancies.”

Lines 378-382: “Taking into account that its upregulation is directly related with the amount of neutrophils within the gastric mucosa, which represent an indicator of inflammation activity, we can emphasize the role of miRNA-223 in predicting gastritis severity. Therefore, the overexpression of miRNA-223 might be considered a potential gastric cancer biomarker.”

Lines 397-398: “Most likely, the downregulation of the five aforementioned miRNAs defines their usefulness in detecting chronic gastric inflammation.”

Thus, by this letter and by the attached revised manuscript of our original manuscript, we hope to have fulfilled all the observations and recommendations made by the Reviewers.

Thank you for your time and consideration.

On behalf of all authors of this work,

Yours sincerely,

Lecturer Lorena Elena Meliț, MD, PhD

Departament of Pediatrics I, “George Emil Palade ”University of Medicine, Pharmacy, Sciences and Technology Târgu Mures, 38 Gh. Marinescu St., 540139, Târgu Mures, Romania. Phone: +40-742-984744. Fax: +40-265-211098, e-mail: lory_chimista89@yahoo.com

Round 2

Reviewer 1 Report

I would like to thank the authors because they have addressed my comments well and thoroughly. The figures they made are not only an added value to their manuscript, they are also very clear and pleasing to the eye. The legend of each figure should contain a small piece of text explaining what is depicted on the figure and guide the readers through the figure. So please, add that to the final manuscript. Next, I would like to ask the authors to check the manuscript for double spaces and delete those. Concerning Line 164: please change transcript to transcribe. I have no additional comments.  

Author Response

January the 26th, 2020

To the Editorial Board of International Journal of Molecular Sciences,

Dear Editor,

Please find attached a revised version of the manuscript entitled: " International Journal of Molecular Sciences," written by Maria Oana Săsăran, Lorena Elena Meliţ and Ecaterina Daniela Dobru, Manuscript ijms-1090559

Firstly, we thank very much the reviewers for their valuable comments and suggestions in order to improve our paper.

Following the reviewers’ concerns and observations, we made some modifications to the initial version of our manuscript, which we described in detail, according to their recommendations, highlighting them in yellow in the attached manuscript. The number of the lines mentioned below can be found in the  “Manuscript SMO Changes accepted_MicroARN_IJMS_ 26.01.2021” uploaded document.

Reviewer 1

I would like to thank the authors because they have addressed my comments well and thoroughly. The figures they made are not only an added value to their manuscript, they are also very clear and pleasing to the eye. The legend of each figure should contain a small piece of text explaining what is depicted on the figure and guide the readers through the figure. So please, add that to the final manuscript. Next, I would like to ask the authors to check the manuscript for double spaces and delete those. Concerning Line 164: please change transcript to transcribe. I have no additional comments.  

First of all, we would like to thank you for your constructive comments and suggestions, which helped us substantially improve the quality of our manuscript and create the three figures which help the reader better visualize the described pathways.

An explanatory paragraph has been added with each figure legend, at the end of the manuscript:

Figure 1; lines 736-740: “H. pylori presents a unique helicoidal shape which ensures effective colonization of the gastric mucosa. Flagella are essential for its mobility, whereas LPS is responsible for triggering host immune response. The urease, the main enzyme of the bacterium, is located inside the cytoplasm, where it maintains a constant alkaline pH, which facilitates efficient survival of the pathogen inside the gastric acid environment. VacA and CagA are two of H. pylori’s main virulence factors. Once delivered to host cells, these two entities can mediate inflammatory pathways leading to carcinogenesis.”

Figure 2; lines 746-750: “IL-8 expression is upregulated as a result of H. pylori infection and plays a central role in the development of gastric cancer. This cytokine is released after activation of NF-kB- and AP-1- dependent pathways. Its chemotactic properties for neutrophils will eventually facilitate formation of a tetrameric network (CD11b/CD18/neutrophil/ICAM-1), which will induce ROS release through activation of NOX-1. ROS damage epithelial cells and stimulate MAPKs, which further enhance AP-a transcription and, subsequently, IL-8 release. The cytokine is also responsible for VEGF release, with angiogenic properties. “

Figure 3: lines 757-761: “NF-kB dependent pathways are activated as a result of TLR-2/TLR-4 and LPS interactions and stimulate release of pro-inflammatory cytokines, including IL-12 and IL-1β, responsible for differentiation of Th1 and Th17 cells. The three main virulence factors of H. pylori positively impact release of inflammatory cytokines, VacA being the only one which is not directly involved in TLR-LPS interaction. Illustrated miRNAs seem to be able to both negatively modulate NF-kB expression and release of inflammatory cytokines. Furthermore, a few of them are downregulated by CagA (i.e. let-7b) and by NF-kB (i.e. miRNA-155).”

The manuscript was thoroughly checked for double spaces and those were deleted. Furthermore, the word “transcript” from line 164 was replaced with “transcribe”.

Thus, by this letter and by the attached revised manuscript of our original manuscript, we hope to have fulfilled all the observations and recommendations made by the Reviewers.

Thank you for your time and consideration.

On behalf of all authors of this work,

Yours sincerely,

Lecturer Lorena Elena Meliț, MD, PhD

Departament of Pediatrics I, “George Emil Palade ”University of Medicine, Pharmacy, Sciences and Technology Târgu Mures, 38 Gh. Marinescu St., 540139, Târgu Mures, Romania. Phone: +40-742-984744. Fax: +40-265-211098, e-mail: lory_chimista89@yahoo.com

Reviewer 2 Report

Authors have answered my questions adequately.

Author Response

January the 26th, 2021

Dear Reviewer,

Please find attached a revised version of the manuscript entitled: " International Journal of Molecular Sciences," written by Maria Oana Săsăran, Lorena Elena Meliţ and Ecaterina Daniela Dobru, Manuscript ijms-1090559

Firstly, we thank very much the reviewers for their valuable comments and suggestions in order to improve our paper.

Following the reviewers’ concerns and observations, we made some modifications to the initial version of our manuscript, which we described in detail, according to their recommendations, highlighting them in yellow in the attached manuscript. The number of the lines mentioned below can be found in the  “Manuscript SMO Changes accepted_MicroARN_IJMS_ 25.01.2021” uploaded document.

Reviewer 2

Authors have answered my questions adequately.

Thank you so much for your positive feedback. We are happy that we managed to answer your questions appropriately and to fulfill your requirements.

Thus, by this letter and by the attached revised manuscript of our original manuscript, we hope to have fulfilled all the observations and recommendations made by the Reviewers.

Thank you for your time and consideration.

On behalf of all authors of this work,

Yours sincerely,

Lecturer Lorena Elena Meliț, MD, PhD

Departament of Pediatrics I, “George Emil Palade ”University of Medicine, Pharmacy, Sciences and Technology Târgu Mures, 38 Gh. Marinescu St., 540139, Târgu Mures, Romania. Phone: +40-742-984744. Fax: +40-265-211098, e-mail: lory_chimista89@yahoo.com